# Pulse Root Ideotype for Water Stress in Temperate Cropping System

**DOI:** 10.3390/plants10040692

**Published:** 2021-04-03

**Authors:** Shiwangni Rao, Roger Armstrong, Viridiana Silva-Perez, Abeya T. Tefera, Garry M. Rosewarne

**Affiliations:** Agriculture Victoria Research, Horsham, VIC 3400, Australia; Roger.Armstrong@agriculture.vic.gov.au (R.A.); viri.silva-perez@ecodev.vic.gov.au (V.S.-P.); abeya.tefera@ecodev.vic.gov.au (A.T.T.); garry.rosewarne@agriculture.vic.gov.au (G.M.R.)

**Keywords:** dicotyledon, indeterminacy, tolerance, avoidance, dryland cropping, constraints, morphology, phenology, legumes

## Abstract

Pulses are a key component of crop production systems in Southern Australia due to their rotational benefits and potential profit margins. However, cultivation in temperate cropping systems such as that of Southern Australia is limited by low soil water availability and subsoil constraints. This limitation of soil water is compounded by the irregular rainfall, resulting in the absence of plant available water at depth. An increase in the productivity of key pulses and expansion into environments and soil types traditionally considered marginal for their growth will require improved use of the limited soil water and adaptation to sub soil constrains. Roots serve as the interface between soil constraints and the whole plant. Changes in root system architecture (RSA) can be utilised as an adaptive strategy in achieving yield potential under limited rainfall, heterogenous distribution of resources and other soil-based constraints. The existing literature has identified a “‘Steep, Deep and Cheap” root ideotype as a preferred RSA. However, this idiotype is not efficient in a temperate system where plant available water is limited at depth. In addition, this root ideotype and other root architectural studies have focused on cereal crops, which have different structures and growth patterns to pulses due to their monocotyledonous nature and determinant growth habit. The paucity of pulse-specific root architectural studies warrants further investigations into pulse RSA, which should be combined with an examination of the existing variability of known genetic traits so as to develop strategies to alleviate production constraints through either tolerance or avoidance mechanisms. This review proposes a new model of root system architecture of “Wide, Shallow and Fine” roots based on pulse roots in temperate cropping systems. The proposed ideotype has, in addition to other root traits, a root density concentrated in the upper soil layers to capture in-season rainfall before it is lost due to evaporation. The review highlights the potential to achieve this in key pulse crops including chickpea, lentil, faba bean, field pea and lupin. Where possible, comparisons to determinate crops such as cereals have also been made. The review identifies the key root traits that have shown a degree of adaptation via tolerance or avoidance to water stress and documents the current known variability that exists in and amongst pulse crops setting priorities for future research.

## 1. Introduction

Pulses are a key seed crop both in Australia and globally due to their production value, their ability to increase soil nitrogen through fixation, forming a disease break in crop rotations, and for leaving residual moisture deeper in the soil profile for use by subsequent, deep-rooted crops [1,2]. In Australia, the annual pulse crop was valued at AUD $1.8 million in 2018 [3]. Furthermore, approximately 3 million tonnes of nitrogen are fixed annually via root nodules in pulse crops, which translates into approximately AUD $4 billion [4]. If constraints are addressed, predicted pulse production can reach 4.2 million tonnes, with a combined commodity value and farm system benefit of AUD 2 billion [3].

Considerable research effort over several decades has been expended on physiological improvements of crop adaptation to water stress [5,6]. A relatively recent line of enquiry has targeted features of the root system architecture (RSA) that specifically improve access to soil water and nutrients. The root system architecture of monocots has been thoroughly characterised in terminal drought conditions such as in that of the Northern Australian cropping system. However, there is comparatively little information available on dicotyledon roots, especially pulses. In contrast to cereals, pulses are an indeterminate crop; hence, they differ significantly in their interactions between phenology, root development and response to seasonal variation [3]. Understanding the RSA of pulses and exploring the genetic variability that favours constraint tolerance or avoidance when combined with appropriate management strategies could provide a direction to breeders to improve the productivity of pulses under challenging climate and soil conditions.

Recently, a “Steep, Deep and Cheap” ideotype [7] has been proposed to increase soil water and nitrogen acquisition in an effective manner in monocotyledons. Steep root angles and thick long laterals allow deep roots to reach soil water and nitrate at depth, whilst a structure with few laterals, large cortical cells and aerenchyma reduces the photosynthate demand on the plant (Figure 1a). The focus on deeper roots is supported by research that has demonstrated the value of water stored deep in the soil profile to seed yield in environments characterised by terminal drought [8], particularly in sub-tropical regions, where crops rely on water stored during summer fallows and “opportunity cropping” [9]. However, deep roots may not be as important in major cropping regions in temperate Southern Australia. Here, rain predominantly falls during the cropping season, heavy clay soils limit the ability of water to permeate deeper down the profile, and fallows are limited to summer when limited recharge occurs. This can result in a lack of stored subsoil water and deeper roots may not provide a benefit during water-deficit conditions. Furthermore, many cropping soils in the southern region contain soil physicochemical constraints such as high boron, sodicity, salinity, high soil strength/poor soil structure and nutrient deficiencies that potentially restrict root growth and access to soil water and nutrients [10] (Appendix A). Thus, there is a need to investigate root traits specific to water stress adaption in the dryland cropping systems of Southern Australia. 

Studies to date on pulse root system architecture (RSA) have identified key traits such as rooting depth, lateral root intensity, root length distribution, root angle and root diameter that have the potential to increase plant adaptation to water stress [11,12,13]. Ramamoorthy et al. [14] investigated some of these traits in in chickpeas and observed that genotypes with higher root length density closer to the soil surface, with greater root dry weight at depth, showed the best tolerance to drought. Similarly, Kashiwagi et al. [15] observed that genotypes with deeper roots and more soil exploration capability perform better under drought stress. Other studies, such as those by Gorim and Vandenberg [16] and Idrissi et al. [17], have examined root trait diversity in selected germplasms and found significant genetic variability in root traits and their response to drought. However, it is difficult to compare the value of various traits across existing studies due to the range of different methods that have been used. Consolidating root traits would provide breeders and agronomists with a new perspective in approaching the development of climate-ready pulse crops, especially when attempting to improve crop adaptation to specific environments.

As a result of the literature reviewed, proposed in this article is an alternative root architecture to the previously proposed “Steep, Deep and Cheap” ideotype [7], namely a “Wide, Shallow and Fine” root that is more aligned to the soil and climate conditions of the dryland cropping systems of Southeastern Australia (Figure 1b). This proposed root ideotype has been developed from reviewing the root system architecture of the major pulses cultivated in the dryland cropping systems of Australia, including chickpea, lentil, faba bean, lupin and field pea. The proposed ideotype of pulse root architecture suited to temperate dryland cultivation also has application to other areas in the world where pulses are cultivated in similar conditions, as in the origin fertile crescent that covers Western Iran, Iraq, Jordan and Israel to Southeast Turkey [18]. In this review, root traits are discussed in relation to drought conditions, focusing more on intermitted drought due to its presences in the dryland cropping system. Furthermore, this review summarises pulse root system architecture and contributes to the identification of knowledge gaps to enable a better understanding of how specific pulse root traits can assist in improving pulse productivity in the range of soil types, seasonal conditions and agronomic management systems that dominate Southeastern Australia. 

## 2. Ideotype Context and Traits

### 2.1. Soil, Climate and Constraints

Pulse cultivation in the temperate regions of Australia with low- to mid-rainfall zones is mainly undertaken in dryland systems. Rainfall in the low-rainfall zone is <350 mm, medium 350–550, and in the high-rainfall zone, it is >550 mm. Pulses in temperate Australia are typically sown in late autumn and experience cool wet winters, with terminal drought often ending the season in early summer [19]. Existing cultivars of these crops are best adapted to alkaline soils and mostly grown on Vertosols, Sodosols and Calcarosols with a high-clay subsoil. Faba bean appears to be able to tolerate more acidic soil and waterlogging in high-rainfall areas. However, most other pulses cannot and are more limited to higher soil pH in medium- and low-rainfall areas (Appendix A). Topsoils can be both clay and loam, although sands become increasingly dominant in some of the lower rainfall areas. Environmental conditions in Southern Australia drylands result in a range of physicochemical and nutrient constraints, including water stress, subsoil compaction, salinity, sodicity, boron, extremes in pH and nitrogen and phosphorous deficiencies. 

Despite the numerous abiotic and biotic constraints to pulse productivity in dryland cropping systems, the principal abiotic constraint is drought [10,20]. Dryland cropping systems in the medium- and low-rainfall zones of Australia typically experience unpredictable rainfall, often in insufficient amounts to replenish soil water throughout the potential rooting zone of annual crops, whilst in higher-rainfall locations, high precipitation: evaporation rates often lead to waterlogging during winter. Furthermore, the large episodic events that can occur in summer are also inefficient in recharging subsoil water storage due to high runoff and evaporation rates [19,21]. Verburg, McBeath, Armstrong, Tavakkoli, Wilhelm, Mclaughlin, Haling, Richardson, Mason, Kirkegaard and Sandral [21] found that, in certain areas, insufficient and irregular rainfall resulted in low soil water, starting at depths of only 10 cm in the soil profile.

In the case of dryland cropping systems in Southern Australia, where subsoil water can be limited due to unpredictable rainfall and continuous cropping, traits associated with developing deep roots may provide little benefit in most temperate cropping systems and may even reduce productivity as excess energy is used in developing roots that provide little value. However, critical knowledge can be extrapolated from these traits and applied to root system architecture suited to alternative seasonal rainfall patterns and soil types. 

### 2.2. A New Root Ideotype—Wide, Shallow and Fine 

Given the propensity for many pulse cropping regions to experience moderate in-season droughts, a new root ideotype that combines a range of root architectural traits is proposed. This ideotype has a root length density that is concentrated in the upper soil layers to capture in-season rainfall before it is lost due to evaporation. This high root length density would be achieved through more hypocotyl roots, a wide root angle allowing for lateral root growth in a shallow soil profile, thin lateral roots and thin xylem diameter with a high number of root hairs to increase surface area to volume ratio. Lynch [7] proposed cheap roots in terms of construction costs, by going deep quickly, and this was important to ensure a balance on the photosynthate demand on the plant. Shallow roots are inherently cheap to produce as there is no wastage on extensive root production whilst searching for water; shallow roots are already at the optimal position for water and nutrient uptake in ephemeral rainfall systems. A high water use efficiency (WUE)/transpiration efficiency and harvest index are also preferred for the optimal extraction of soil moisture and to convert it effectively to biomass and yield. Furthermore, the development of early root vigour, aimed at establishing extensive colonisation of upper soil layers when soil water is available and root penetration is easier, would likely benefit productivity. Early vigour above ground can also help to trap soil moisture by reducing evaporative losses. Early root and shoot vigour would accelerate the accumulation of shoot biomass early in the season, before water deficiency reaches limiting levels, and could act in concert with efficient translocation of assimilation to seed later in the season. The proposed ideotype would also help in competitiveness against weeds. Ramamoorthy, Lakshmanan, Upadhyaya, Vadez and Varshney [14] have also proposed a more profuse root length density in the topsoil and few thick roots at depth as adaption traits for water stress. The possession of high hereditability and plasticity in this ideotype is also important, so that in years with above-average rainfall, yield potential is still able to be fully achieved. Section 3, Section 4 and Section 5 further discuss the key root system architecture traits that have been reported in pulses and have contributed to the understanding and development of the concept of the “Wide, Shallow and Fine” root ideotype. 

## 3. Morphological Traits and Their Responses to Water Stress

### 3.1. Root System Differences between Cereals and Pulses

The RSA is the spatial configuration of roots within a root system and encompasses aspects of the morphology, topology and distribution of complex root systems [13]. Roots represent an expensive investment by plants compared to shoots in terms of assimilation usage [22,23]. However, this investment can show a positive return via increased productivity when a critical resource such as water or nitrogen (N) is in short supply.

The dicotyledon root system varies significantly from monocotyledons due to the configuration of xylem and phloem, the absence of defined pith and the presence of cambium, which allows dicotyledons to form woody stems [24]. The main root structures in dicotyledons consist of the taproot, lateral and hypocotyl roots, whereas monocotyledons have a crown, bracket, seminal and lateral roots. In dicotyledons, taproots are typically the longest and thickest root structures, with lateral roots branch off at varying intensities down its length. By comparison, in monocotyledons, the initial root structure, the radical, becomes indistinguishable from the later forming seminal roots. Thus, there is no predominant single root in monocotyledons (Figure 1). Hypocotyl roots are similar to the crown root structure in monocotyledons, emerging due to genetic influence [25] and contributing to mineral and water absorption [7]. In addition, unlike cereals (monocotyledons), pulses have a greater level of indeterminacy and have the ability to reflower late in the season if late rains occur. Another major difference to cereals is the ability of pulses to form symbiotic root nodules with rhizobium bacteria, fixing atmospheric nitrogen (N) and reducing the requirement to seek mobile N from the soil. This review focuses on examining tap and lateral root traits; thus, root nodules have not been reviewed. In addition, the degree of root branching is also not included in this review as investigation of this trait in pluses could not be found; however, it can be understood through root length, surface area and weight distribution.

### 3.2. Taproot Length and Rooting Depth

Taproot length and rooting depth are closely correlated unless the taproot is damaged or if the lateral roots extend beyond the taproot. Some of the earlier studies, as discussed by Gregory [26], observed pulses such as field pea, lentil and faba bean to have shallow roots in comparison to cereals in similar environments. Chickpea has been identified as often having deeper roots than lentil, field pea or faba bean [26] (Appendix A). 

Rooting depth has been highlighted as a key trait for drought adaptation, as deep roots can access more water during drought. Increases in rooting depth under water stress conditions have been observed in multiple crops, including pulses, and are considered an avoidance adaptation [17,27,28]. Due to the relationship between soil strength and soil moisture content [29], root growth ceases if soil moisture falls below critical potential. Although deep roots are an important adaptation to drought conditions, they may not be suitable for every form of water deficit. In the dryland cropping systems, deep root cereals can deplete water to a depth of well below what a subsequent pulse crop could reach [1,2]. If subsoil water is not replenished over summer, deeper pulse rooting depth could be a negative resource investment for the plant and for subsequent crops as there may be little stored subsoil water to access. Deeper roots also require a combination of specific root traits, including rooting angle, root diameter and root length [7]. A deeper rooting depth also requires a suitable environment of low soil bulk density, soil strength and sufficient pore size to allow ease of growth down the soil profile. Lucas et al. [30] and Han et al. [31] found that, in soil with pre-existing macropores, root growth was promoted, resulting in a highly porous rhizosphere, more root length and deeper roots. However, where roots were creating macropores as they grew, with no pre-existing macropores, there was induced compaction. Generally, rooting depth has low heritability and is an environment-specific trait [32,33].

### 3.3. Lateral Root Number Intensity and Density in the Soil Profile

Lateral roots are a major structure of root system architecture, responsible for water and nutrient absorption. The number of lateral roots is strongly influenced by genetic make-up, resulting in high heritability and consistency between seasons [34]—an important consideration in selection by crop breeders. Lateral roots in lentil have been documented to emerge at a depth of 2–4 cm and 11 first-order roots were recorded in a four-day growth period with an average diameter of 0.2–0.3 mm [35]. In pulses, chickpea appears to have the highest number of lateral roots (Appendix A), with a maximum documented lateral root number of 764 [36].

Lateral roots are regarded as a potential drought tolerance trait due to the correlation of lateral root number with water-stress-induced wilting [17]. Idrissi, Houasli, Udupa, De Keyser, Van Damme and De Riek [17] showed that lateral root numbers tend to decrease in water-stressed environments, which correlates with decreases in shoot and root biomass, chlorophyll content and root surface area and an increase in the root: shoot ratio. Liu, Gan, Bueckert and Van Rees [12] showed a similar response in chickpea. However, in lentils, they observed an increase in lateral root number under water-stress conditions. Field peas showed no significant difference. The distribution of lateral root number appears to be dependent on genetic influence and has been identified as a heritable trait; however, it lacks plasticity, as observed by Liu, Gan, Bueckert and Van Rees [12] in field pea and [37] in faba bean, where it did not vary significantly under water stress compared to a well-watered environment. Apart from increasing or decreasing the lateral root numbers, depending on the dynamic of the stress, plants have shown other modifications, such as altering the thickness of roots [14,17].

### 3.4. Root Diameter

Root diameter determines the radial hydraulic pressure in lateral roots and the compounding effect of radial and axial pressure on the taproot [38]. The axial pressure, in turn, governs the ability of roots to elongate and penetrate soils [39]. In the field, root diameter varies according to the highly heterogeneous soil environment; a higher level of resistance is encountered by roots in soils with low porosity, high soil strength and high bulk density. Roots can accumulate biomass by creating thicker roots with high hydraulic pressure to buckle and push through soil peds [39] or they can respond by reducing root diameter to squeeze through soil pore spaces [40]. The diameter of roots grown in medium with minimal resistance decreases from proximal to the distal tip of the root [38,41].

Root diameter is a heritable trait and has been used to differentiate varieties as well as their adaptive response to constraints [14,25]. Pulse crops have been observed to have thicker-diameter roots in comparison to cereal and oilseed roots [12]. This allows pulses to create macropores, creating the potential for these crops to grow better than cereal and oilseed crops in soils with low porosity and high soil strength. 

Root diameter has been observed to decrease under water-stressed conditions [14,17]. This decrease in diameter results in thin roots with a greater surface area per unit of dry matter, allowing more surface area for water absorption as well as the better acquisition of immobile resources such as PO_4_. However, thicker roots are often associated with increased penetration of high-soil-strength soils [40]. Therefore, a combination of well-branched thin and thick roots would be ideal for environments in dryland cropping systems where crops are reliant on subsoil water. This structure would also benefit nitrogen and phosphorous uptake [42].

#### Xylem Vessel Diameter

The diameter of xylem vessels determines axial pressure on roots and water use efficiency [11,43]. In wheat, increased yield was observed in a cultivar with a thinner xylem diameter that restricted water use early in the season whilst leaving enough water for the grain-filling period [44]. Richards and Passioura [44] successfully developed wheat varieties bred for narrow xylem that showed a 3–11% yield advantage under water-stressed conditions but not under well-watered conditions. Richards and Passioura [45] also found that xylem diameter was a more heritable trait in comparison to the number of seminal root axes in wheat. 

Reduced water uptake as a result of decreased xylem capacity—from the number of xylem tubes and/or their diameter—is a significant breeding consideration. If pulse varieties lack sufficient plasticity to adjust xylem diameter under different water conditions, then, under well-watered conditions, a yield penalty may occur. 

## 4. Physiological Traits and Their Response to Water Stress

### 4.1. Vertical Stratification 

#### 4.1.1. Rooting Density

Root length density (RLD) generally decreases down the soil profile, with pulses having a lower overall density in comparison to cereals in the top 50 cm of the profile. However, below 50 cm, wheat rapidly decreases in density to a level matching that of pulses [12]. Pulses have indeterminate root growth, which provides a level of plasticity as they can grow roots as required depending on intrinsic physiology or extrinsic environmental factors. Kashiwagi et al. (2006), observed that root length density in chickpea is quite plastic in its response to environmental factors over time. Appendix A provides a comparison of RLD in pulses as root length (cm) per cm^3^ of soil.

At the late flowering stage, field pea had the highest RLD above 30 cm depth, followed by lentil and chickpea (Liu et al 2011b). At pod fill stage, 60% of the RLD measures were in the top 15 cm of soil (Manshadi et al., 1998) and, at maturity, soybean had significantly higher root weight density in the top 40 cm of soil compared to chickpea and faba bean [46]. 

Under both well-watered and drought conditions, RLD had a positive correlation to yield, especially at greater depths [14,47]. Fluctuations were observed in RLD during the growth period as it is required to accommodate grain filling under variable soil water availability. A more profuse RLD at the surface and high root dry weight at depths was favoured as an adaptive trait for drought tolerance [14].

An increase in RLD observed under water stress suggests an avoidance response as plants grow more laterals to explore and extract soil moisture, thus avoiding water stress conditions, while a decrease in RLD is a tolerance adaptation. Several studies have identified RLD decreases under water-stress conditions in chickpea and field pea, suggesting plasticity-based tolerance mechanisms [12,46,48]. Both the tolerance and avoidance response occur in the top 15–30 cm of the soil profile.

#### 4.1.2. Root Mass and Root:Shoot Ratio

Under well-watered conditions, increases in total root biomass have been observed in different lines that do not influence yield, but under water-limiting conditions, increased root biomass can be a major factor as it contributes to water absorption [49,50]. In well-watered conditions, root biomass decreases markedly with increasing soil depth across multiple pulse crops. Chickpea and lentil have been observed to invest almost a third of root biomass in the top 0–15 cm soil layer (Appendix A) [15,51]. The 15–30 cm zone is the second-largest root biomass investment and corresponds to the section of the soil profile where water uptake is greatest [15]. These two sections have been observed to have the highest genotypic variation, as below 30 cm, root biomass declines steeply and genotypic variations become insignificant [15,36,51]. 

Under water-stress conditions, root biomass has been documented to increase, decrease or have no significant change [14,16,17,46,50,52]. An increase in root biomass suggests an avoidance mechanism as plants can uptake nutrients and water from the soil profile and in-season rainfall [14,16,17]. A decrease in biomass suggests a tolerance mechanism because plants stop investing in roots and conserve resources for investment later in seed development [52]. A lack of significant change in root biomass under water stress demonstrates a lack of plasticity or use of other plant traits in actively adapting to environmental stimuli [46,50]. These variations in plant root and shoot biomass response to water stress could be accounted for by interspecies or intraspecies variation and the degree of water deficit. 

### 4.2. Surface Area

Root surface area is critical in determining the amount of area available for water and nutrient absorption. The most active regions of water and nutrient absorption provided by the root surface are the root hairs and the region 20–30 cm behind the root tip, as older roots become suberised and lose the capacity to function as absorption tissue [38,53]. 

Under field conditions, chickpea, field pea and lentil have shown a lower root surface area per unit mass of root than wheat but higher than the oilseed crops of canola, flax and mustard (Liu et al 2011). Amongst pulses, field pea was observed to have a higher surface area in the top 0–20 cm of soil, followed by chickpea, then lentil, whilst, deeper in the soil at 40–80 cm, chickpea had a higher surface area, followed by field pea and lentil (Liu et al 2011). Similar results were observed in another study comparing chickpea, field pea and soybean [46]. Thus, chickpea and field pea, with high surface area and low root weight, have a finer root system than soybean (Appendix A). 

Under water-stress conditions, pulses appear to respond by either restricting the growth of roots completely and decreasing root surface area and biomass [54] or by increasing the surface area and decreasing root biomass [46]. The restricted growth under water deficit could potentially be a tolerance-adaptive mechanism to conserve resources. Alternatively, the densifying of root systems by increasing surface area can be considered an avoidance mechanism as lighter or finer roots allow for more water absorption and reduced water loss. These responses have been found to vary both between [12,46] and within species [54]. Both responses to water stress would be beneficial in dryland cropping. However, densifying of roots in the topsoil potentially has greater benefits as it would maximise the area for water extraction in environments dominated by in-season rainfall, thereby preventing soil water loss via evaporation.

### 4.3. Root Angle

Root angle is a heritable genotypic trait defined as the angle formed between the lateral and taproot, main axis or the horizontal soil layer [25,41,55]. This review will adopt the first definition. Root angle is regulated by polar auxin in the root tip, which creates a negative gravitropic response [56,57]. Seedling and mature root angles are highly correlated. Measurement of seedling root angle in seedlings grown on germination/filter paper or petri dishes has become a means of high-throughput screening [25,41,55]. 

Deep roots as a result of narrow root angle that allow access to stored subsoil water have been shown to be a useful adaptive trait in the northern grain cultivation region and under drought conditions in Queensland, Australia [55]. Deep roots have also been associated with better nitrogen acquisition as the bio-available nitrate is quite mobile and travels down the soil profile with water [58]. 

A wider root angle typically results in shallow roots, which can be beneficial for dryland in-season rainfall in combination with agronomic practices, such as reduced tillage and residue retention. A wider root angle may also enhance phosphorus (P) acquisition as P is usually insoluble in its organic form and is relatively immobile in soil, resulting in greater stratification near the soil surface [59]. Hypocotyl roots with wider root angles growing near the soil surface have been identified as improving tolerance to low soil P [60,61]. Hypocotyl roots could also potentially aid in water uptake during intermittent drought before it is lost to evaporation. 

### 4.4. Water Movement in Pulses 

#### 4.4.1. Water Use Efficiency (WUE) in Pulses

Water use efficiency (WUE) is an expression of the amount of product (grain or biomass) produced per unit of water. It can be expressed either in terms of actual water used (mm) as transpiration efficiency (m^−2^ mm^−1^) or in terms of the amount of rainfall received (assuming there was no net change in soil water balance). Water use efficiency is a key metric for assessing adaption in water-constrained environments as it is positively correlated with grain yield in most dryland cropping systems [62]. A comparison of WUE across pulse, cereal and oilseed crops indicates that pulses such as pea, lentil and chickpea generally have higher WUE than oilseed crops but lower than cereals, at 4.08, 5.5–7 and 3.64 kg ha^−1^ mm^−1^, respectively [2]. Wild lentil varieties that were putatively adapted to drought conditions generally had greater WUE under water-limiting conditions than cultivated varieties, except for one cultivated lentil genotype, Eston [16]. Figure 2 provides a summary of root architectural traits that have been associated with water uptake, movement and use efficiency.

#### 4.4.2. Water Movements in Pulse Roots

The uptake and transportation of water in plants is a prerequisite for good water use efficiency. Varney and Canny [63] found that, in maize plants, the water movement or flux between root branches and root axis was very similar, at an average flux of 5 µL h^−1^ cm^−2^. However, Ahmed et al. [64] identified greater water uptake in maize crown roots compared to seminal and lateral roots, whilst water uptake in lupin was predominately by lateral roots, where radial fluxes increased, moving from the root tip to the proximal region, regardless of soil depth. The total radial flux of roots in the upper zone (2–9 cm) was higher than in the lower zone (18–27 cm) in 21-day lupin seedlings. However, these observed differences in flux measurements are likely to be seen in more distal roots as the roots mature and experience decreased radial flux and increased axial conductivity [38]. The diffusion permeability of cortical tissue was consistent along the root, whilst the permeability of the endodermis decreased from 2–3 cm to 10–12 cm. Similar studies are warranted for hypocotyl roots so as to understand the role that they play in plant water uptake and adaptation to abiotic constraints.

#### 4.4.3. Pulse Root Water Extraction Potential

Pulse crops have been observed to have a lower rate of water extraction in comparison to wheat [1]. Of the pulses examined, chickpea was observed to have the highest amount of water extraction (approximately 12% higher) in the 60–100 cm soil depth range compared to lentil and pea. The shallow depth of water extraction by pulses such as lentil compared to wheat results in higher residual water for subsequent crops in the rotation following a pulse crop [1]. Campbell, Zentner, Basnyat, Wang, Selles, McConkey, Gan and Cutforth [1] also observed that lentil used the same amount of soil water as wheat, despite having a shallower root system and lower biomass. The ability of lentil to extract water equivalent to wheat suggests that water potential is greatly reduced in the region of the soil occupied by lentil but leaves water below this region of occupation. 

The ability of plants to extract water is influenced by both soil properties and the plants [65]. The permanent wilting point is classically set at a matrix potential of −1500 KPa. Wang et al (2012), using soil water extraction (SWE), found that, at maturity, chickpea was able to extract water and reduce soil water potential to a greater extent while having a similar WUE to other pulses under both low- and medium-rainfall environments. Chickpea was closely followed by faba bean, dry bean, dry pea and lentil. Wang et al. [66] also observed that the increase in water availability during high-rainfall years reduced WUE especially in pea, chickpea and faba bean, suggesting that WUE is a plastic response. It has also been determined to be a highly heritable and polygenic trait [45]. Examination of the general trends of water extraction rather than the actual amounts extracted would be beneficial as these are strongly influenced by rainfall. For instance, Zaman-Allah, Jenkinson and Vadez [50] observed that in the 38 days after sowing (DAS), tolerant varieties of chickpea under water-stressed conditions extracted less water during the early growth stages, leaving water in the soil profile for later stages. Drought-sensitive plants exhibited the same level of water extraction in water-stressed environments as the control of the well-watered environment. Post 38 DAS, the tolerant varieties were able to extract more water than the sensitive ones, allowing better grain fill and quality.

## 5. Phenological Traits and Their Response to Water Stress

### Root Phenology

In wheat plants, root growth has been mapped parallel to shoot growth, with leaf number and tillering corresponding to root growth and roots reaching maximum growth at early flowering [67]. In oilseeds such as canola and mustard, maximum growth occurs at late flowering, whilst in flax, it occurs at the late pod stage. In pulses, root growth is positively correlated with shoot growth and appears to halt at the late flowering/flat pod stage, after which it senesces through pod fill and desiccation [67]. 

Indeterminate growth in pulses may result in a secondary root flush if late rains occur after an intermittent drought and can result in significant increases in grain yield compared to determinate crops in some situations. This aspect of the root phenology of pulses has been relatively underutilised as a water stress adaptation. Figure 3 provides an illustration of potential root phenology and yield timing in pulses compiled from an understanding of the literature on pulse root growth and the works of Liu, Gan, Bueckert and Van Rees [67] and Rawson and Macpherson [68].

While reduced or irregular rainfall is the main contributing factor to water stress in plants, root growth may not always be synchronised with changes in soil water potential or location. A potentially important breeding and agronomic consideration is that plants with roots that have vigorous growth early can absorb more water in the initial growth stage and build more biomass than plants that have low early root vigour [69]. Early root vigour can prove beneficial if the genotype has a high harvest index and good WUE. However, if the plant has a low harvest index and poor WUE, early vigour roots can deplete soil moisture and induce water stress during the grain-filling stage, resulting in yield loss. Seedling vigour has already been identified as a drought adaption trait in Mediterranean environments [32]. Early root vigour can also be an attractive adaption trait for dryland cropping with Mediterranean environments as water is usually available early in the season and depletes towards the end of the season, thereby maximising growth and assimilate accumulation when there is little water stress, which can be converted to yield later in the season. Crops can send roots deep into the subsoil during early growth stages when there is plenty of water in the soil profile but, as the season progresses (and rainfall decreases), the impact of salinity on plant growth becomes greater under increasing soil water potential (due to increasing osmotic potential). Therefore, a prerequisite for the use of early vigour in terms of rooting depth is the absence of salinity and boron in the relevant soil profiles.

## 6. Research Priorities

It has been strongly argued that much of the current success in improving the grain yield of crops under water stress has centred on “avoidance” strategies rather than “tolerance” [70]. Examination of the existing literature on root traits has provided a conceptual framework for targeted, constraint-related root research into tolerance and avoidance. However, research on water stress needs to expand beyond terminal drought conditions to address other forms of water deficit encountered in pulse cultivation, such as in dryland cropping. 

Figure 4 identifies the various roots traits that need to be investigated to provide breeders with information on traits that can help to improve yield potential in constrained environments. These traits are the fundamental building blocks of a climate-ready ideotype. A number of traits that have the potential to increase plant adaptation to water stress in temperate climates in the absence of stored subsoil water have been identified in the proposed ideotype, including rooting depth, lateral root intensity, root length distribution, root angle, root diameter, water extraction and use efficiency, harvest index and phenology. However, further requirements of the ideotype need to be identified regarding the physical and chemical constraints of soils, such as soil compaction and impacts of boron toxicity, salinity and sodicity. The interrelationships between many of the identified constraints, root architecture and yield have been poorly explored. Research designed to assess the impacts of multiple constraints in multiple locations will provide agronomists and breeders with the requisite information to prioritise trait selection in new varieties.

Furthermore, unlike cereals, pulse roots are indeterminate and, as shown in this review of the current literature, the significant lack of information regarding the relationship between root architecture and yield in pulse crops has limited our understanding of individual pulse crops and their root responses. Future research on the traits identified in Figure 4 should prioritise understanding the extent of pulse root indeterminacy and longevity. Research should also be focused on root phenology, how it synchronises with above-ground phenology and its implication on yield. 

Research into the genetic diversity and quantitative trait loci (QTLs) is becoming increasingly important in RSA, with a number of QTLs being identified for deep rooting, root angle, root length and drought tolerance that have been listed and discussed in detail in a recent review by Siddiqui et al. [71]. However, all these investigations and trait detection have been undertaken on cereals such as rice, wheat, barley, maize and sorghum. Similar studies need to be conducted in pulses if pulse productivity constraints are to be addressed. As highlight in Appendix A, genetic diversity already exists in pulses. Future research on different varieties and new cultivars alongside investigations into QTLs could aid in the development and refinement of the proposed root ideotype for water-stress conditions. Finally, as more information is accumulated on root traits and response to stimuli, more accurate predictive models can be developed.

## Figures and Tables

**Figure 1 plants-10-00692-f001:**
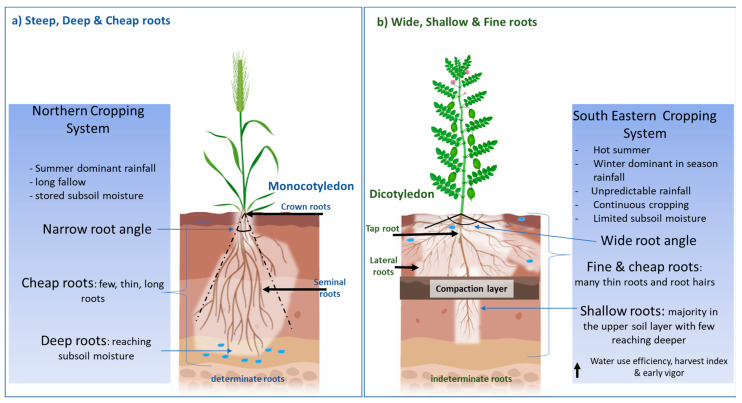
(**a**) The “Steep, Deep and Cheap” root architecture that has been largely investigated and applied to monocotyledons such as maize and wheat cultivated in environments that experience terminal drought with presence of subsoil water. (**b**) The proposed alternative, “Wide, Shallow and Fine” roots, for the southeastern cropping system of Australia, which experiences unpredictable rainfall and absence/unavailability of stored subsoil water.

**Figure 2 plants-10-00692-f002:**
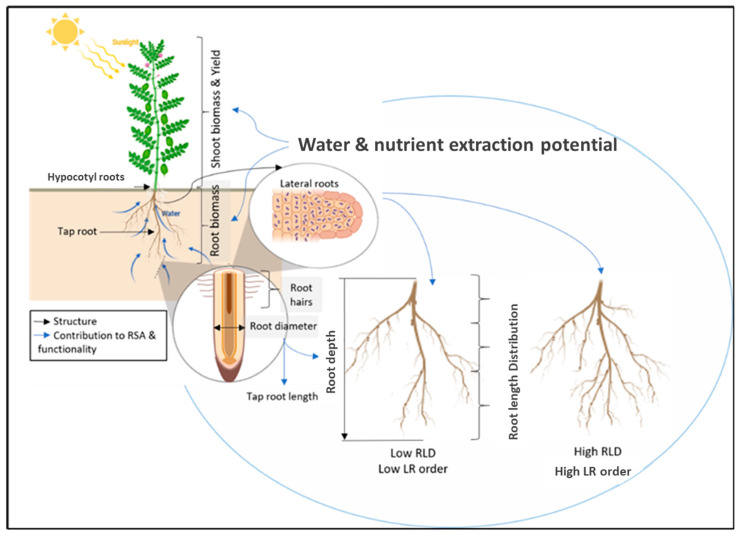
Summary of pulse root architectural traits interrelation and contribution to plant functionality.

**Figure 3 plants-10-00692-f003:**
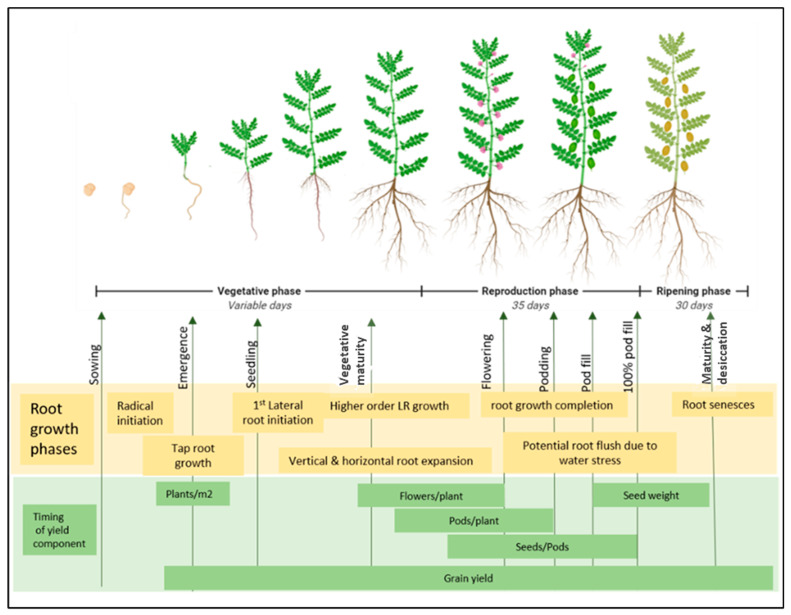
The current understanding of pulse root phenology aligned with the timing of factors affecting yield and yield components. After sowing, the days to maturity of pulse crops are variable depending on temperature, photoperiod and rainfall. The reproductive and ripening phase can last for approximately 30 to 35 days from maturity. The first factor for crop yield after sowing is plant emergence, which relies on good taproot growth and establishment. Following taproot, the first-order lateral roots (LR) emerge at the early seedling stage. Early vigour at a seedling stage is critical for biomass accumulation and harvest index. From seedling to flowering, roots expand rapidly, growing deeper into the soil and densifying with the emergence of higher-order laterals. Root growth contributes to nutrient and water acquisition, which feeds into the number of successful flowers that transition to seeds. During pod filling, soil moisture levels can be greatly reduced and root flush potential in pulses may allow the last surge of root growth to extract soil moisture to a greater extent.

**Figure 4 plants-10-00692-f004:**
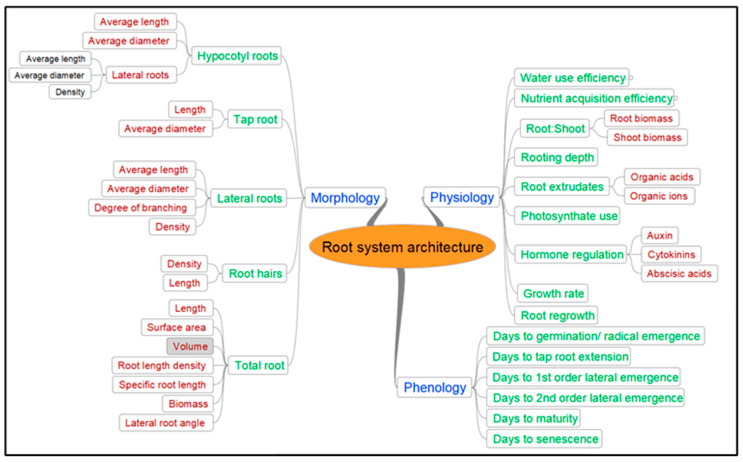
Spheres of root system architecture highlighting morphological, physiological traits and potential pulse root phenology.

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
