# Peer review of "Pulse Root Ideotype for Water Stress in Temperate Cropping System"

_plants, 2021, doi:10.3390/plants10040692_

Round 1

Reviewer 1 Report

In general, this review manuscript is well organized and had potential scientific importance to the field. It is well written except for minor comments mainly for the Abstract.

Abstract:

I find it not organized. Please rewrite the abstract in a way the purpose of this review clearly stated and organized. Then state the suggested model.

-Line 16: We propose..... please avoid using personal pronouns and apply this rule throughout the manuscript.

2. Ideotype context and traits

2.1. soil, climate and constrains

Line 113-117( The large episodic ......during winter) please provide a citation

3. Morphological traits and their response to water stress

Line 195- 199 (if....and root length) also needs a citation

4-Research Priorities

The author provides several important future research topics, I think it will be more interesting if different varieties and new cultivars were added to the future research to consider the genetic variations and the rule of the genetically modified pulse crops on the suggested models. 

Author Response

Point1. I find it not organized. Please rewrite the abstract in a way the purpose of this review clearly stated and organized. Then state the suggested model.

Response1. The abstract has been revised and now clearly states the purpose of the review followed by the proposed ideotype. L 12-34.

Point 2. Line 16: We propose..... please avoid using personal pronouns and apply this rule throughout the manuscript.

Response 2. The personal pronouns have been removed.

Point 3. Line 113-117( The large episodic ......during winter) please provide a citation.

Response 3. The statement has been referenced. L131. 

Point 4. Line 195- 199 (if....and root length) also needs a citation

Response 4. The statement has been referenced. L213.

Point 5.  Research Priorities

The author provides several important future research topics, I think it will be more interesting if different varieties and new cultivars were added to the future research to consider the genetic variations and the rule of the genetically modified pulse crops on the suggested models. 

Response 5. The suggestion by the reviewer has been included in the research priorities section in L524-534.

Reviewer 2 Report

Climate change has increased the occurrence of extreme weather patterns leading to a significant decrease in crop production, and hence global food security. Currently, water scarcity is a major constraint in the production of pulses. Root system architecture (RSA) is a vital agronomic and developmental trait, which plays an important role in plant adaptation and productivity under water-limited areas. 

The present review is well written and summarizes key root traits associated with enhanced water-stress tolerance in pulses. I recommend publication of review after the addition of a minor component in the revised manuscript:

The authors must include the genetic diversity and analysis of RSA by giving a few examples of pulse crops.

Author Response

Point 1. The authors must include the genetic diversity and analysis of RSA by giving a few examples of pulse crops.

Response 1. The authors acknowledge the suggestion by the reviewer and inline with comments from reviewer 1 a paragraph on root trait genetic diversity has been included please see L 524-534. The authors would also like to highlight that in addition to various pulse crops being discussed in the manuscript supplementary table 1 also lists all the pulse crops that have been investigated to date and the diversity of their measured root traits.